# Ecological modelling: A computational analysis of air pollution discourses in English print media of India and Pakistan

**Sana Rabbani, Fasih Ahmed** *

Department of Humanities, COMSATS University Islamabad, Islamabad, Pakistan

* ahmadfasih33@gmail.com

## Abstract

The present study investigates air pollution dynamics through newspaper discourse in India and Pakistan, where both countries rank among the top five countries affected by air pollution. The study focused on newspaper discourses over almost two decades (2005–2023). The study applied Latent Dirichlet Allocation (LDA), a robust algorithm for analyzing the large text corpus. The study underpinned Computational Grounded Theory, which relies on the fact that computation is a way to reveal the hidden meanings beyond the text. The LDA-generated topics reveal that both countries face the toxicological effects of air pollution on health. The primary topics extracted through LDA revolve around discourses related to vehicular emissions, industrial emissions, and urbanization. In addition, the control measures taken by both countries relate to emission standards. The study also has implications for policymakers and planners considering these directions to control air pollution.

**Data Availability Statement:** The data has been shared. Please find below the doi of the data shared through figshare. 10.6084/m9. figshare.26969395

## 1 Introduction

Air pollution is an environmental concern worldwide, threatening human health, the natural ecosystem, and the economy [1,2]. As a global concern, it has sparked numerous debates on Sustainable Development Goals (SDGs) to address the issue on local, national, and international levels, particularly in developing countries [3]. SDGs aim to improve air quality by reducing pollutant emissions, providing efficient renewables, and designing policies for energy, climate, transport, agriculture, biodiversity, and other contributors to environmental pollution. The idea of SDG is based on a mutual acknowledgment at the global level that intense air pollution will not only lead to a high risk for disease and low level of social welfare but also impose immeasurable negative impacts on sustainable development in the long run [4].

Air pollution is the contamination of the atmosphere by the emission and transmission of gaseous, liquid, or solid wastes that can harm human health, quality of life, and natural functioning of the ecosystem, reduce visibility, or produce undesirable odors [5]. Pollution levels have progressively increased globally over the past two decades due to rapid urbanization and a rise in industrial, commercial, biogenic, and human activities [6]. This has led to complex chemical processes that have increased the atmospheric concentration of pollutants such as

**Funding:** The author(s) received no specific funding for this work.

**Competing interests:** The authors have declared that no competing interests exist.

carbon monoxide, nitrogen oxides, sulfur oxides, non-methane volatile compounds, hazardous metals, ammonia, and free radicals [7]. However, the rising levels of air pollution are a priority concern, especially in developing countries, due to the lack of stringent pollution control measures, rapidly growing population and limited resources to address pollution, and lack of access to healthcare [8].

This research examines how the issue of air pollution in the two developing countries, Pakistan and India, has been linguistically represented in newspaper reports over the past 18 years. According to the World Health Organization WHO (2019), despite being a global crisis, air pollution has become a growing threat, specifically in the metropolitan cities of developing South Asian countries in the past two decades. Most of the population of these countries is affected by this phenomenon. South Asian countries, such as Pakistan and India, historically remain among the world's top five most polluted countries [9]. Since 2005, pollution levels in Pakistan and India have risen exponentially, crossing the highest thresholds of pollutant concentrations worldwide [10]. IQAir's world major-city air quality index (2022) ranks India's capital, Delhi, as the first and Pakistan's city, Lahore, as the second most polluted city globally. Delhi's Air Quality Index (AQI) in December 2022 is reported as 321, which is currently 54.1 times the WHO annual air quality guideline value, i.e., 0–100, putting it in the 'hazardous (301+)' bracket of the US AQI. Similarly, Pakistan's city, Lahore, is placed second among all the cities worldwide, with the AQI reported as 299 classified as 'Very Unhealthy (201–300)' (IQAir, 2022). Therefore, it has become imperative to address the issue of air pollution in Pakistan and India.

The present study follows the timeframe from 2005 to 2023. The rationale for the timeframe selection is to focus on the timeline from when the Particulate Matter (PM) 2.5 concentrations in each of the countries crossed the threshold of 60 micrograms per cubic meter ($\mu g/m^3$) (12 times the recommended value) recorded as the highest in the history of the two countries. The historical data show a drastic increase in the concentration of pollution-causing PM from 2005 onwards. Fig 1 shows a timeline of the years in which the issue of air pollution gained maximum traction in Pakistan and India.

The current study investigates the issue of air pollution from the linguistics perspective by analyzing newspaper discourses regarding air pollution. Newspapers are a source to bring to the fore the hazards, challenges, and policy measures brought by air pollution. Print media discourse also represents "a significant aspect of political representation," which includes participation from various social bodies such as government representatives, industrial associations, scientific groups, or environmental organizations that can lead to public opinion formation [11]. Hence, newspapers maintain the role of a significant "interpretive system" in modern societies to raise public awareness (Peter Peters et al., 2008).

The newspaper discourse on air pollution in the context of developing countries such as Pakistan and India, which rank among the top five countries affected by air pollution for the past two decades, is an issue yet to be explored from the lens of linguistics. Therefore, the current study contributes by examining an environmental phenomenon in the most affected countries over nearly two decades to gain a holistic insight into the hazards, challenges, and national adaptation strategies brought about by the issue.

## 2 Literature review

Air pollution is an environmental crisis and a leading worldwide health risk factor [12,13]. With outdoor air pollution reportedly accounting for around 8% of all global deaths [14]. Despite being a global health hazard, air pollution is of particular concern in densely populated developing countries like Pakistan and India, which are among the top five most polluted countries worldwide [9]. A plethora of epidemiological research has associated poor air quality

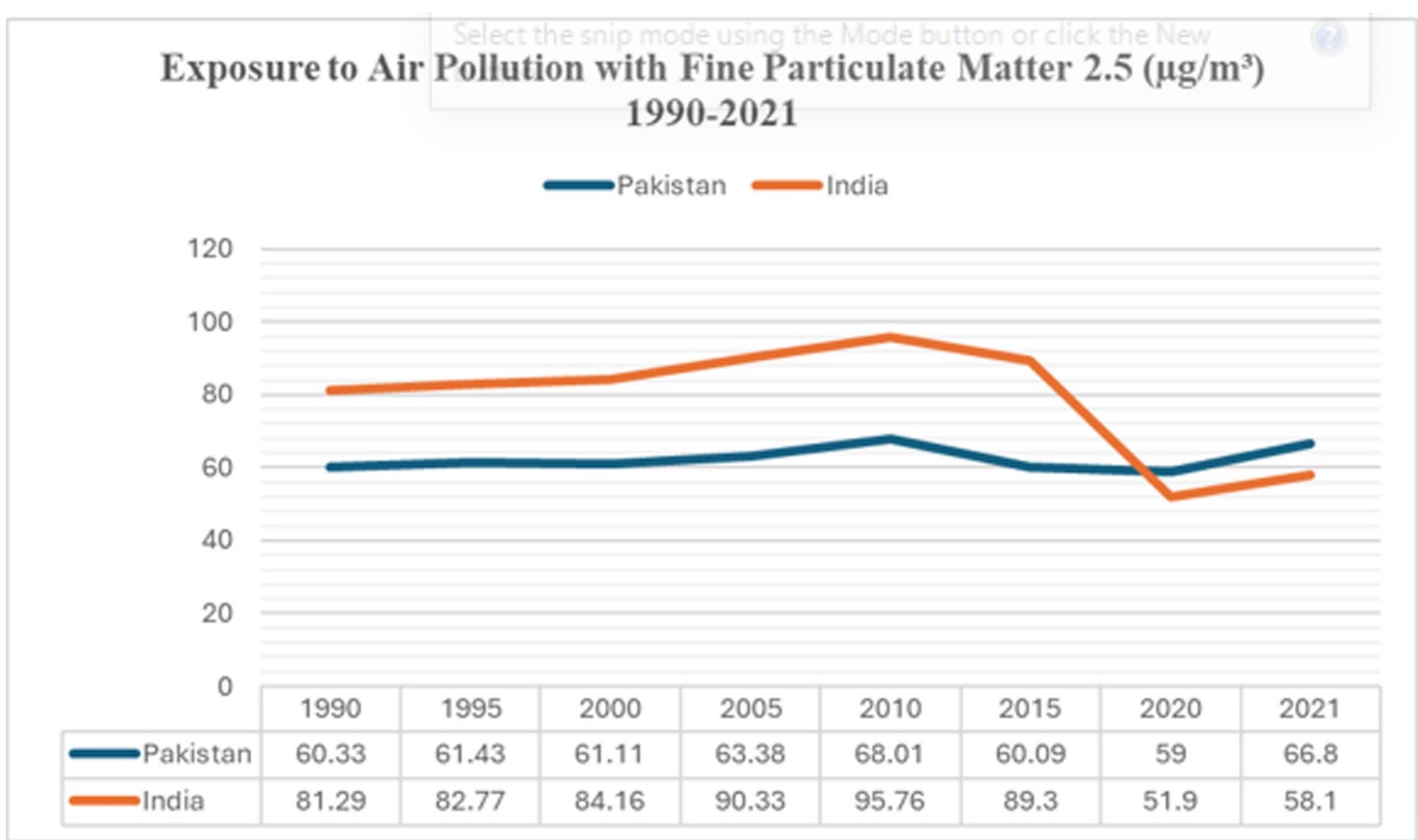

**Fig 1. Exposure of air pollution particles in India and Pakistan [9].**

with a variety of health conditions and rising mortality rates in developing countries [8,15]. Air pollution accounts for 6.5 million mortalities annually, with two-thirds of the mortalities occurring in developing countries, including Pakistan and India [16]. In India, more than one million people die annually from exposure to household air pollution alone [14]. Likewise, mortality rates in Pakistan are alarmingly high due to increased exposure to air pollution, mainly particulate matter [17].

The dissemination of information through news sources can increase awareness, alter views of environmental issues, such as air pollution, and assist individuals in adopting risk-reducing behaviors [18]. Reportedly, the severity of air pollution-induced risks is more pronounced, especially in developing countries like Pakistan and India, which lack adequate preventative, warning, and protection systems [17]. Numerous studies exploring newspaper coverage of environmental issues have focused mainly on high-income (developed) countries [19]. The context of developing countries such as Pakistan, India, and Bangladesh, which ranked among the top five countries affected by air pollution for the past decade, have gained less attention of researchers, undertaking media sources for the analysis. However, recently, a slight shift in the research trends has initiated research on the media coverage of climate change in middle- and low-income countries [5]. The few exceptions, such as the climate change-related newspaper report analysis in India from the year 2004 to 2009 and in Bangladesh from the year 2004 to 2006, fail to capture the adverse implications caused by the alarming increase in levels of air pollution since 2009 [20,21].

Ecolinguistics, a sub-field of Applied Linguistics, examines "the general patterns of language that influence how people think about and treat the world" [22]. Ecological language analysis, as a critical approach in ecolinguistics, examines the text in connection to the environment to reveal the underlying discourses and evaluate the degree to which they serve the desired environmental objectives [2,23]. It studies the relationship between language and ecology to represent how discourses in media, society, or politics positively or negatively influence people's understanding and treatment of the environment [24,25]. The symbiosis between language and ecology gained rationalistic emphasis by the claim that "the destruction of species, pollution, and the like are not just problems for the biologists and physicists but are concerns of the applied linguistic community" [26]. Since then, numerous researchers have embraced Ecolinguistic discourse analysis focused on the intersection of language, ecology, and society [23,27–30]. The present research builds on the newspaper corpus's computational ecological discourse analysis in this context.

The past two decades have witnessed an increase in climate-related discourse, necessitating the application of automated text analysis techniques to examine vast corpora of environment-related texts [31–34]. The framework of the present research allows the use of the computer-assisted Topic Modeling (TM) tool Latent Dirichlet Allocation (LDA) in processing massive corpora of texts and discovering relevant underlying semantic patterns while also calling attention to the inclusion of discourse analytic methodologies in interpreting the results produced by such tools [35]. Social science research has applied LDA to investigate islamophobia and anti-feminism [36], child abuse [34], policymaking [37,38], economy and UN assembly speeches [39,40]. Automated-text approaches to discourse analysis have the potential to analyze a large sample of texts efficiently with empirical evidence for testing research assumptions and provide results that cannot be achieved via manual analysis of a small sample of texts [41] . In addition, these approaches eliminate the researcher's bias in data selection, interpretation and increase the replicability of the outcome.

## 3 Research methodology

The present study employs a mixed-methods research design using both qualitative and quantitative approaches to analyze a corpus of Pakistani and Indian full-length newspaper articles reporting on air pollution. The data were initially analyzed through computation, which is a quantitative technique, and then this data was analyzed to find out the grounded realities in data using a qualitative approach. Hence, the study adopts qualitative and quantitative approaches to gain insight into air pollution.

### 3.1 LDA

The research employs LDA to uncover the text's latent or hidden thematic patterns. LDA assumes that every document in the corpus consists of a mixture of latent word patterns called topics and calculates their probability distribution over the documents [35]. In this regard, each document can be viewed as a probability distribution across a range of topics, and the topics that have the highest probabilities are the ones that best describe the document. The word clusters, "topics," represent latent associations between words across a corpus and are identified via Latent Dirichlet Allocation [42,43].

Automated text analysis assists in what were once labor-intensive and time-consuming manual practices through TM approaches that facilitate the analysis of the corpus [44]. This research employed TM, an unsupervised technique used to simplify data and discover a set of coherent and substantively meaningful topics on the newspaper coverage of air pollution [38].

The present study applied topic modeling using the Gensim library in Python. This library is helpful in conducting topic modeling, specifically LDA. One more issue is that the number of topics is not predetermined. However, refining the model may determine the number of topics. In the case of the present study, we initially set 20 number of topics. The observation of the topics and the relevant themes revealed that some of the topics were redundant. Some of the keywords were observed as redundant or noisy. In order to remove the noise, the researchers kept decreasing the number of topics until they reached a reasonable number of topics that depicted the relevant themes.

## 3.2 Sample

The corpus generated for this research is a large textual dataset extracted from Pakistan and India's newspapers. The full-text news articles about air pollution from Pakistan and India's leading newspapers indexed across the LexisNexis database *Nexis Uni* within the timeline (1st January 2005 - 31st December 2023) were extracted using comprehensive keyword search operators explained in Table 1.

When considering newspaper circulation in both countries, it is important to observe that newspaper circulation has increased significantly in Pakistan and India throughout the years mentioned in Figs 2 and 3.

The criteria for the selection of the newspaper outlet for each of the countries was based on five factors: a) daily publication, b) universal national coverage, c) large circulation, d) reputable journalistic standards, and e) highest news index on air pollution. Following these criteria, the news articles on air pollution in Pakistan were extracted from the four mainstream Pakistani newspapers: *Dawn*, *The Nation*, *The News International*, and *Right Vision News*. After removing the group duplicates, 1,236 full-text news articles from all four newspapers were assembled to account for the corpus of Pakistan. The news articles on air pollution in India were extracted from the two mainstream Indian newspapers, *Hindustan Times* and *The Times of India*. The two selected newspapers yielded a total of 4,872 news articles.

The process of identifying and gathering relevant news articles on air pollution from the LexisNexis database involved using comprehensive search operators. These search operators shown in Table 1 were constructed by combining various keywords chosen based on a thorough review of previous literature to ensure the search results were highly relevant.

The number of newspaper articles and sources varies country-wise. The first variation is based on sources. In the case of the Indian corpus, the data were retrieved from two sources, whereas in the case of the Pakistani corpus, the data were retrieved from four sources. The reason for this is that, in the case of Pakistan, the existing reporting articles regarding air pollution were not enough for topic modeling. Hence, considering the limited number of articles, more sources were consulted to understand the dynamics of air pollution in Pakistan. Due to variations in the size of the data country-wise, the topic modeling has been conducted separately to mitigate the margin of sample size difference. Moreover, for extracting the topics, the search operators were kept the same in the case of both countries to maintain uniformity in the data.

**Table 1. The detail of corpus generation countries, sources, and operators.**

| Corpus Generation | | | | | |
| --- | --- | --- | --- | --- | --- |
| Search Operators: Air Pollution AND Haze OR Smog OR Disease | | | | | |
| Pakistani Corpus | | | | Indian Corpus | |
| Dawn | The Nation | The News International | Right Vision News | Hindustan Times | The Times of India |
| 236 | 260 | 282 | 459 | 2,086 | 2,786 |

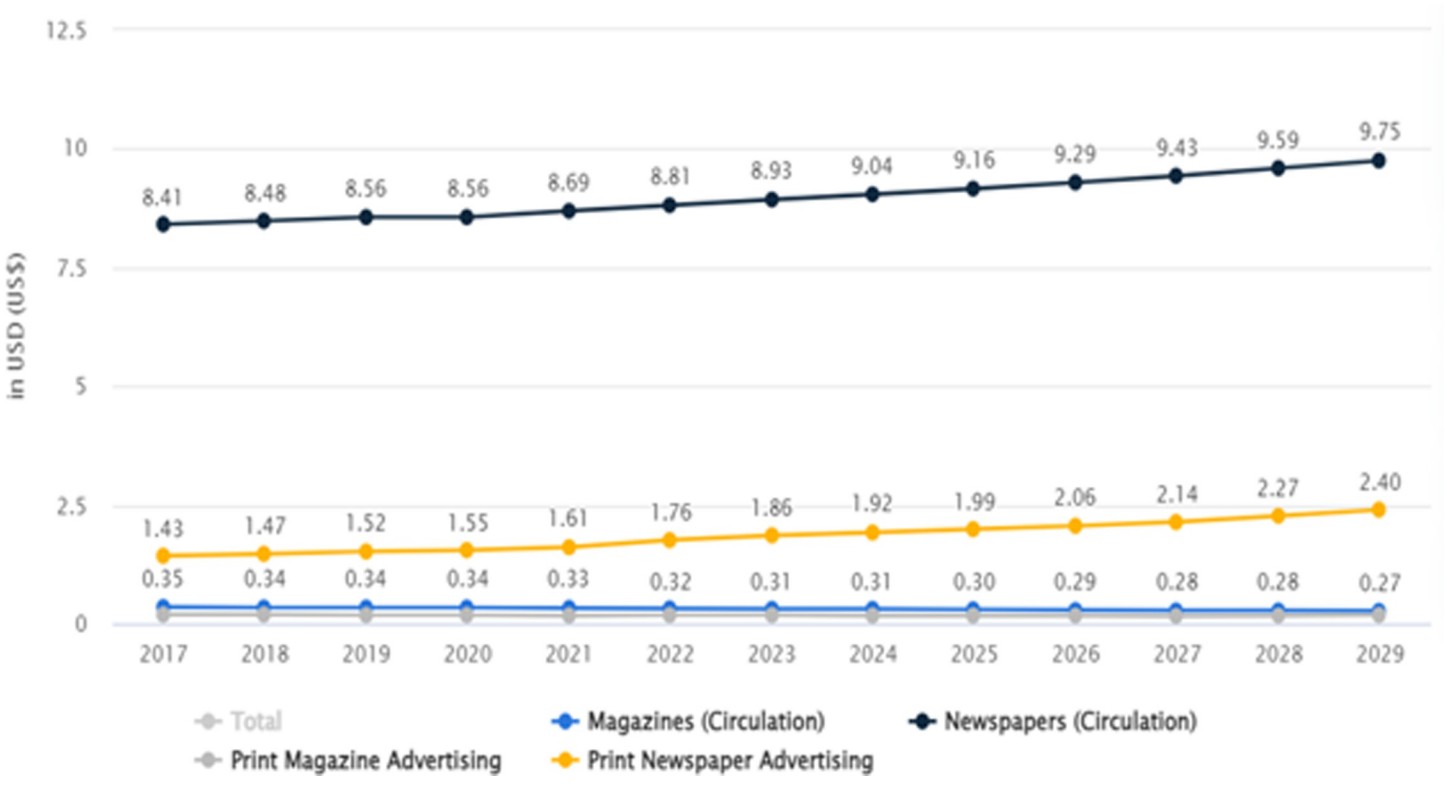

**Fig 2. Newspaper circulation in Pakistan ([45]).**

### 3.3 Data filtration

Standard data preprocessing was completed before building a term document matrix from which topic models were estimated. The preprocessing phase was conducted using Python programming. Data Filtration steps ensure the relevant data is extracted against each news article. The first step in pre-processing is tokenization, which segments the large body of the text into sentences that are further split or tokenized into smaller units in the form of words. At this stage, the tokenized units undergo parts of speech tagging and also take into account bigrams such as air_pollution, climate_change and trigrams such as black_carbon_emissions in addition to unigrams which contributes to improving the topic quality [43]. The second step followed stemming and lemmatization, in which the words are reduced to their meaningful base form called Lemma. For example, the word forms pollution, pollutant, and polluting are reduced to the root form pollute. The third step followed the procedures to clean out the noisy data, such as removing punctuations, numbers, common stopwords, URLs, hashtags, and person names and converting all text to lowercase. All the above processes were conducted using different packages from libraries in Python, such as NLTK, for tokenization and lemmatization.

### 3.4 Data validation

Topic Modeling is a probabilistic technique that involves a variety of tradeoffs and judgments on the part of the researcher to ensure the semantic validity of the identified topics [45]. For this purpose, the researchers used qualitative and quantitative validation techniques, including semantic, statistical, and predictive validation, to confirm the validity of the topic models [43].

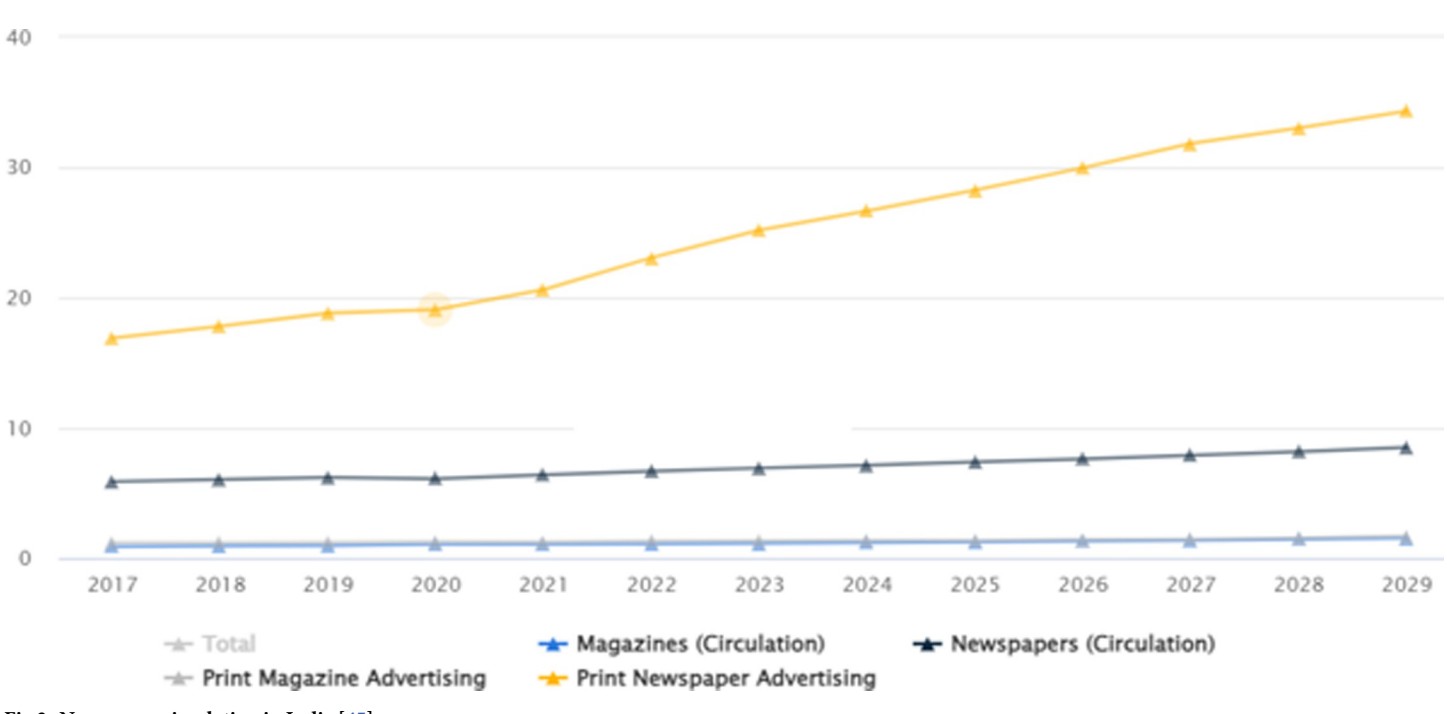

**Fig 3. Newspaper circulation in India** [45].

Firstly, the topic models generated from each country's corpus were evaluated separately keeping in view the semantic meaningfulness and intelligibility level. Another important measure for ensuring the "relevance of topics" and avoiding redundancy was the calculation of Term Frequency and Inverse Document Frequency (TF-IDF) weights, which was achieved using Scikit-learn, an open-source machine learning library for Python. TF-IDFs calculate the importance of a term in a document and in which document it holds the highest importance. This evaluation helped filter out the topic models of common adjectives, general terms, and irrelevant topics. The selected model for Pakistan's corpus comprised ten topics that fulfilled the validity requirements. Similarly, the model chosen for India's corpus consisted of nine topics.

### 3.5 Computational grounded theory

The research underpinned Computational Grounded Theory (CGT), which combines the power of computational methods with human expertise as a theoretical framework (Nelson, 2020). CGT combines the benefits of computational methods with the benefits of the human ability to interpret, established by the classical method of GT. The combination forms a framework to derive reproducible, valid, reliable, and more transparent results. The methodological framework of CGT follows three steps: pattern detection step, pattern refinement step, and pattern confirmation step [46]. The pattern detection step uses computer techniques to simplify the text into smaller units to reveal hidden patterns in the data. The pattern refinement step involves a re-engagement with the text via computationally guided deep reading. The pattern confirmation step uses further computational techniques to assess the reliability and validity of the inductively identified and refined patterns.

## 4 Data analysis

The ten topics identified from Pakistan's corpus are assigned themes through a rigorous examination of the corpus to understand the identified topics in the context of the corpus (Isoaho et al., 2021). Table 2 shows the ten topics' keywords and the weights mentioned for each keyword, left with the assigned labels for each topic.

The first set of keywords in Table 2 reveals discourse regarding the status of measures undertaken by the government to deal with the issue of air pollution in the country. The Pakistan Environmental Protection Agency (PEPA) (0.017) has made progress (0.017) by not granting permission (0.064) for setting up new coal (0.033) power plant projects (0.038) in the province of Sindh (0.027). The keywords in this topic indicate that progress (0.017) has been made at the provincial (0.015) level to deal with the rising levels of air pollution in the country.

The second set of keywords emphasizes the adverse effects on citizens' health (0.018) brought on by the country's increasing (0.013) levels (0.022) of air (0.069) pollution. It highlights the need for necessary measures to be taken to protect the health (0.018) of the residents of the city (0.013). The keywords indicate a strong linkage between urban air (0.069) quality (0.011) and human health (0.018). Therefore, the urban environment protection measures should be effectively implemented to ensure citizens' health (0.018) and quality (0.011) of life.

The third set of keywords reflects the official (0.036) efforts in addressing the country's two main drivers of air pollution. The set of keywords in this topic, such as "road" (0.063), "vehicle" (0.056), and "steel" (0.014), indicate vehicular exhaust and industrial activities as the two major contributors to toxic levels of air pollution in Pakistan. While on the other hand, the keywords "official" (0.036)," department" (0.031), "control" (0.021), and "launch" are indicative of the initiatives launched (0.017) by the decision makers to control (0.021) the rising

**Table 2. Topics extracted from Pakistan's corpus using LDA.**

| Topic Keywords | Themes |
|---|---|
| [(0, '0.064*"permission" + 0.038*"project" + 0.033*"coal" + 0.027*"sindh" + "0.017*"pepa" + 0.017*"progress" + 0.016*"direct" + 0.016*"submit" + "0.015*"provincial" + 0.012*"minister"'), | Pollution Control Initiatives |
| (1, '0.074*"pollution" + 0.069*"air" + 0.022*"level" + 0.018*"health" + "0.013*"city" + 0.013*"increase" + 0.013*"cause" + 0.012*"high" + "0.012*"emission" + 0.011*"quality"'), | Health Effects of Urban Air Pollution |
| (2, '0.063*"road" + 0.056*"vehicle" + 0.036*"official" + 0.033*"lahore" + "0.031*"department" + 0.026*"environment" + 0.021*"control" + 0.018*"phase" "+ 0.017*"launch" + 0.014*"steel"'), | Vehicular Emission Control Measures |
| (3, '0.027*"asthma" + 0.019*"case" + 0.017*"query" + 0.016*"people" + "0.016*"add" + 0.012*"citizen" + 0.011*"eye" + 0.010*"suffer" + "0.010*"avoid" + 0.010*"call"'), | Health Hazards |
| (4, '0.044*"country" + 0.022*"health" + 0.021*"global" + 0.021*"world" + "0.019*"need" + 0.016*"energy" + 0.016*"report" + 0.014*"urban" + '0.012*"list" + 0.010*"population"'), | Urbanization |
| (5, '0.048*"climate_change" + 0.028*"brick_kilns" + 0.020*"decrease" + "0.018*"climate" + 0.017*"carbon" + 0.017*"impact" + 0.015*"average" + "0.013*"cold" + 0.013*"country" + 0.012*"flood"'), | Black Carbon Emissions |
| (6, '0.000*"preventative" + 0.000*"proportional" + 0.000*"whop" + '0.000*"impossibility" + 0.000*"conscientious" + 0.000*"shrug" + '0.000*"grass_root" + 0.000*"fauna" + 0.000*"inversely" +0.000*"inculcate"'), | Effects on Agriculture |
| (7, '0.035*"water" + 0.024*"area" + 0.022*"environment" + 0.019*"environmental" "+ 0.013*"waste" + 0.013*"city" + 0.012*"plant" + 0.012*"resident" + "0.010*"tree" + 0.009*"industrial"'), | Industrial Waste |
| (8, '0.028*"government" + 0.020*"city" + 0.020*"noise" + 0.015*"respect" + "0.013*"issue" + 0.012*"ban" + 0.012*"factory" + 0.011*"court" + "0.010*"emit" + 0.010*"come"'), | Mitigation |
| (9, '0.031*"disease" + 0.024*"death" + 0.024*"cancer" + 0.019*"people" + "0.018*"cause" + 0.017*"child" + 0.017*"patient" + 0.013*"health" + '0.013*"risk" + 0.013*"percent"')] | Death Risk |

levels of pollutants in the air. The keywords' validation in the context of the documents in which they appear reveals several initiatives launched (0.017) by the Punjab Environmental (0.026) Protection Department (0.031) to control (0.021) the contributing sources of air pollution in the city of Lahore (0.033).

The fourth set of keywords recounts the consequences of air pollutants on human health. The respiratory system and skin are among the human organs most vulnerable to hazardous air pollutants because the contaminants are either inhaled through the respiratory tract or absorbed through the skin. Therefore, the citizens (0.012) suffer (0.010) from diseases such as asthma (0.027) eye (0.011) irritation, dryness, or breakage of tear film due to air pollution.

The fifth set of keywords identifies urbanization (0.014) in Pakistan as a major source of increasing air pollution and deteriorating health (0.022). Population (0.010) growth and urbanization (0.014) in the country are contributing towards higher energy (0.016) consumption and need (0.019). The reports (0.016) indicate that the rapid increase in urban (0.014) population (0.010) and higher energy (0.016) consumption is giving rise to urban (0.014) sprawl and industrialization. These elements are important contributors to the rising levels of air pollution.

The sixth set of keywords identifies emissions from brick kilns (0.028) as the leading source of spiking air pollution in the country (0.044). Traditional brick kilns (0.028) in Pakistan mainly undergo incomplete combustion of fossil fuels, biomass, and biofuels that emit lethal black carbon (0.017) into the atmosphere as soot. Lethal carbon (0.017) emissions from brick kilns (the finest and most efficient form of particulate matter) in the country (0.044) result in a myriad of adverse environmental impacts, including air pollution and planet-warming impact (0.017) on climate (0.013). Accumulation of black carbon on snow or ice in the country's cold (0.013) regions spikes atmospheric warming and increases the melting rate of glaciers. This increases the risk of glacial lake outburst floods (0.012) in the country (0.013).

The seventh set of keywords suggests that air pollution in Pakistan has damaged the region's flora and fauna (0.000). Air pollutants pose a hazard to flora and fauna by affecting the grass roots (0.000) of plants and by causing fauna (0.000) to relocate. Insects, worms, mollusks, fish, birds, and mammals all interact with their respective environments in distinct ways. By altering the availability and quality of proportional (0.000) food supplies, air pollution degrades the ecosystem or habitat of wildlife. Government must inculcate (0.000) preventative (0.000) measures for the protection of flora and fauna (0.000) in the country.

The eighth set of keywords indicates the nexus between waste (0.013), water (0.035) released by industrial (0.009) units and air pollution. Industrialization (0.009) engenders air pollution in the country by releasing pollutants and contaminants in the form of poisonous gasses and waste (0.013) and water (0.035) into the environment. These pollutants damage the environment (0.022), which in effect deteriorates the health of city (0.013) residents (0.012) located near industrial (0.009) units in the country. Industries must be made responsible for installing waste (0.013) water (0.035) treatment plants (0.012) before releasing their waste into the environment (0.022). This will ensure the availability of clean drinking water (0.035) to the city (0.013) residents (0.012).

The ninth set of keywords relates to the steps taken by the government (0.028) to improve the air quality in the country. The government (0.028) of Pakistan has taken mitigation measures with the intervention of the Supreme Court (0.011) of Pakistan to ban (0.012) the factories (0.012) and industrial units that are non-compliant with the state laws. The government (0.028) has issued (0.013) standard emission guidelines for the factories situated in the city (0.020) residential areas contributing to noise (0.020) and air pollution. The keywords suggest that the Pakistani government (0.028) and the legislative power of the court (0.011) have prioritized taking stringent action against the factors that contribute to pollution in the country.

The tenth set of keywords reveals air pollution as a serious health (0.013) risk (0.013) factor in the incidence of deadly (0.024) diseases (0.031) such as cancer (0.024). The keywords "disease" (0.031), "death" (0.024), and "cancer" (0.024) carry the highest weight which reveals a direct association between air pollution and a rising percentage (0.013) of death (0.024) rate especially among children (0.017).

Fig 4 shows the word clouds of the topics extracted through LDA. This word cloud is based on the weightage of words explained in Table 2. Fig 5 relates to the visualization of the topics extracted through LDA from the corpus based on Pakistani newspapers. Fig 5 also shows the relevancy of topics and the size of the corpus. Lastly, Fig 6 relates to the distribution of topics extracted through LDA. Each topic in Fig 6 carries a different colour. It also visualizes how topics are merged on semantic basis.

In Table 3, the nine topics of India's corpus are assigned labels by carefully revisiting the corpus to understand the identified topics in the corpus context instead of simply relying on

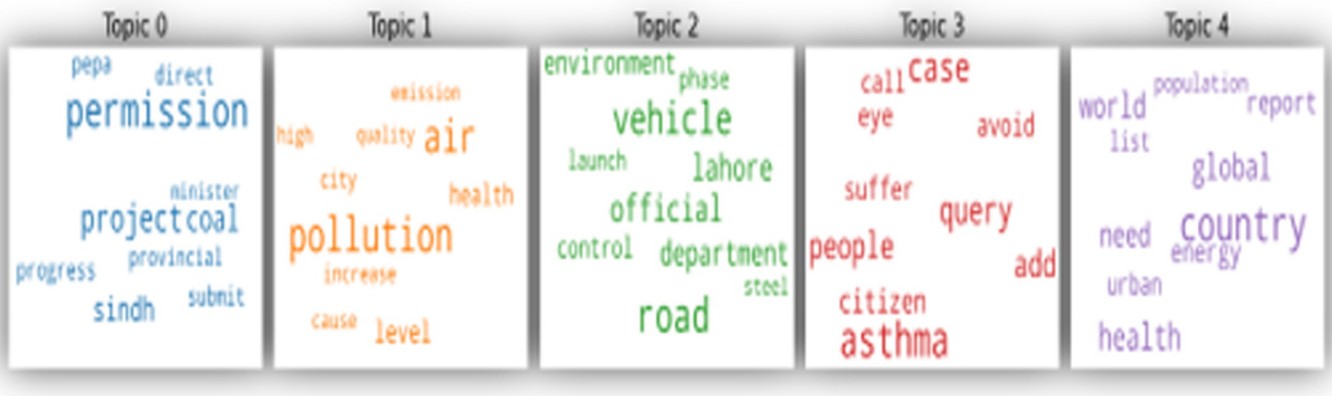

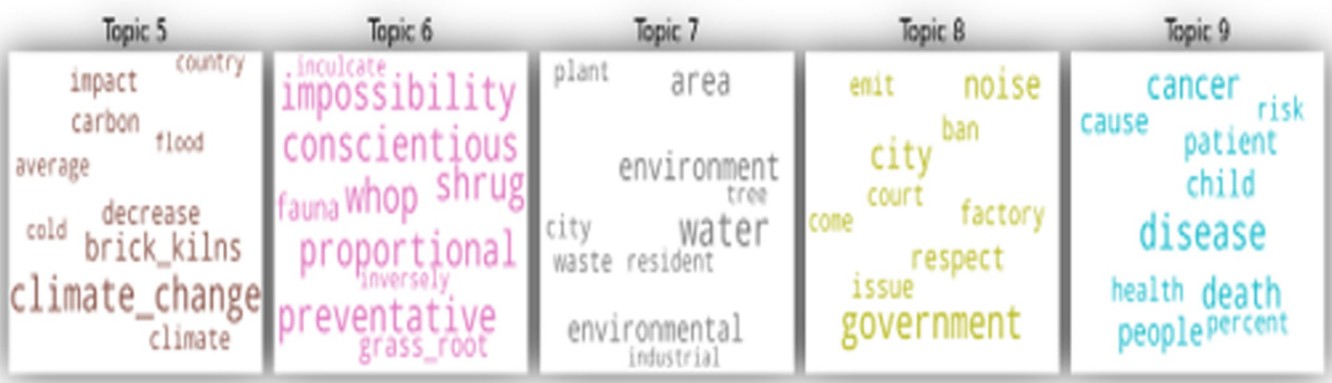

**Fig 4. Word cloud of Pakistan's topics extracted using LDA.**

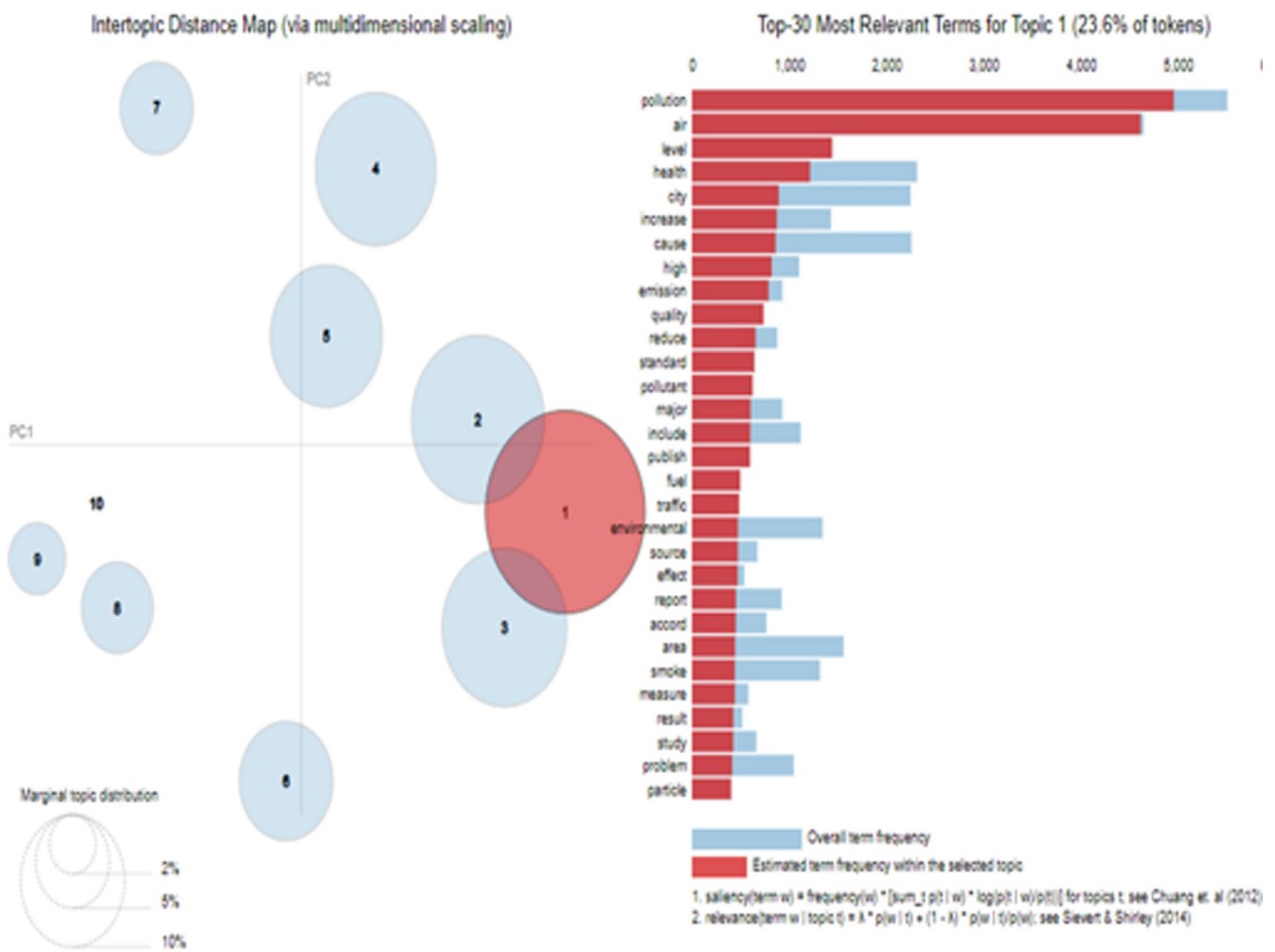

**Fig 5. LDA topic visualization of Pakistan's topics.**

the ten keywords listed for each topic. Table 3 also presents the topics' keywords and a numerical value to the left of each keyword, showing its weight.

The first set of keywords in Table 3 uncover the health (0.032) hazards resulting from exposure (0.012) to air (0.021) pollution (0.028). In this topic, the keyword 'health' carries the highest weight, indicating the impact of air pollution on health. Exposure to air pollution (0.028) is associated with an increased risk (0.013) of diseases (0.024) reported (0.013) around the world (0.018). The severity of the health impacts is further revealed by the keywords: 'death' and 'cancer'. Hence, among the health (0.032) risks (0.013) induced by air pollution (0.028), death (0.014) risk (0.013) due to cancer (0.015) is the most prevalent.

The second set of keywords reveals the risk of developing respiratory (0.013) ailments such as asthma (0.010) and lung (0.017) diseases due to air pollution. The topic suggests that children (0.018) are at the highest risk of suffering from respiratory or lung disorders among all people (0.027). In addition, the respiratory problem (0.012) caused by exposure to air pollution is on the rise significantly during the festival of Diwali (0.010).

The third set of keywords addresses the government (0.025) policies established to reduce air pollution (0.030). The state (0.010) officials' (0.010) implementation of policies to ensure

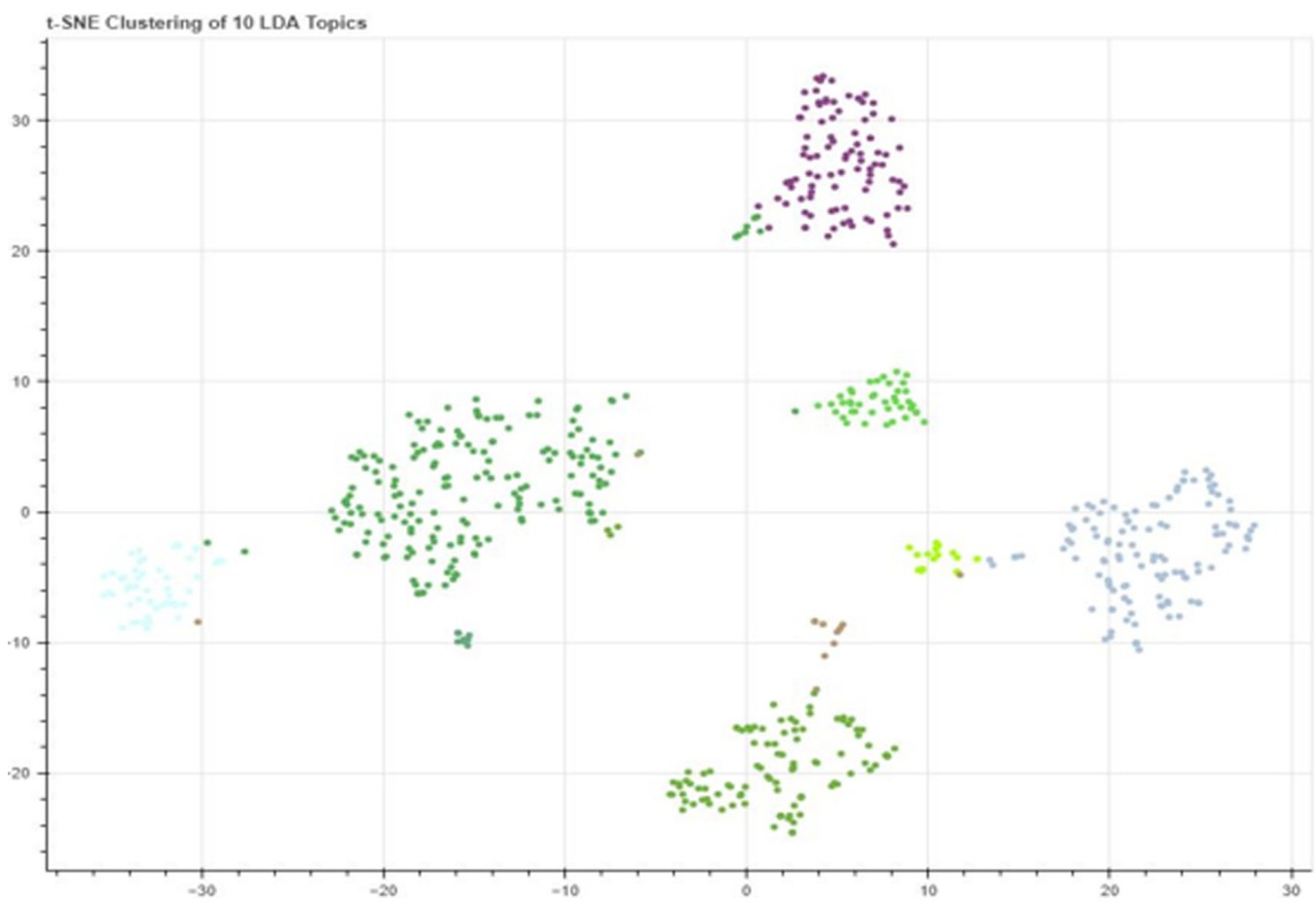

**Fig 6. A t-SNE distribution of topics extracted from the Pakistani Newspapers' corpus.**

quality (0.028) air (0.014) in cities (0.014) is crucial. The policies need to specifically address the issue of stubble burning (0.021), which leads to high air pollution in cities.

The fourth set of keywords brings together a range of interrelated sources of air pollution in India. The construction (0.020) activities performed on the roadside (0.047) act as the primary source of air pollution as the cement dust (0.029) is kept suspended in the air due to the continuous movement of vehicles (0.048) on the road (0.047). The situation is aggravated during the early morning (0.018) hours due to the abrupt spike in traffic (0.016) causing the resuspension of dust particles after the most minor traffic during nighttime. In addition, the continuous vehicular (0.048) movement on roads (0.047) keeps breaking the dust and loose soil particles into fine (0.013) particulates, further intensifying the levels of air pollution. On the other hand, construction (0.020) and vehicular (0.048) exhausts individually appear to be significant contributors to rising air pollution.

The fifth set of keywords reveals air (0.101) pollution (0.072) level as a record (0.016) level (0.066) high in cities (0.045). It indicates that the pollutant (0.025) concentration in the city (0.045) area is reportedly high, specifically with smog (0.013).

The sixth set of keywords emphasizes the need (0.015) for implementing appropriate policies to deal with the issue of rising air pollution. The keyword carrying the highest weight is 'use' (0.024), which indicates harmful human practices as the primary source of increasing air pollution. Therefore, harmful practices such as the use of fuel-powered generators, firecrackers,

**Table 3. Topic models of India's corpus.**

| Topic Keywords | Topic Labels |
|---|---|
| [(0, '0.032*"health" + 0.028*"pollution" + 0.024*"disease" + 0.021*"air" + "0.018*"world" + 0.015*"cancer" + 0.014*"death" + 0.013*"risk" + '0.013*"report" + 0.012*"exposure"'), | Death Risk |
| (1, '0.027*"people" + 0.018*"child" + 0.018*"air" + 0.017*"lung" + 0.013*"cause" "+ 0.013*"respiratory" + 0.012*"problem" + 0.011*"bad" + 0.010*"diwali" + "0.010*"asthma"'), | Respiratory Ailments |
| (2, '0.030*"pollution" + 0.028*"quality" + 0.025*"government" + 0.021*"burn" + "0.014*"air" + 0.014*"city" + 0.010*"official" + 0.010*"permission" + "0.010*"state" + 0.009*"area"'), | Policy Measures |
| (3, '0.048*"vehicle" + 0.047*"road" + 0.029*"dust" + 0.021*"transport" + '0.020*"construction" + 0.018*"morning" + 0.016*"traffic" + 0.014*"bad" + "0.013*"fine" + 0.013*"public"'), | Vehicular and Construction Emissions |
| (4, '0.101*"air" + 0.072*"pollution" + 0.066*"level" + 0.045*"city" + '0.030*"high" + 0.025*"pollutant" + 0.022*"poor" + 0.016*"record" + '0.013*"smog" + 0.011*"polluted"'), | Urban Air Pollution |
| (5, '0.024*"use" + 0.019*"mumbai" + 0.015*"need" + 0.014*"clean" + 0.009*"ban" + "0.009*"try" + 0.008*"people" + 0.008*"cost" + 0.007*"leave" + "0.007*"never"'), | Policy Implementation |
| (6, '0.096*"cracker" + 0.089*"fire" + 0.053*"stubble" + 0.052*"firecracker" + "0.031*"burst" + 0.027*"noise" + 0.020*"acre" + 0.018*"star" + '0.016*"machine" + 0.014*"religious"'), | Firecracker and Stubble Burning |
| (7, '0.039*"film" + 0.036*"plant" + 0.035*"water" + 0.035*"school" + '0.021*"worker" + 0.019*"work" + 0.019*"site" + 0.018*"announcement" + "0.016*"test" + 0.016*"market"'), | Decrease in Quality of Life |
| (8, '0.097*"really" + 0.065*"phone" + 0.065*"cooking" + 0.061*"repair" + "0.041*"insist" + 0.036*"stove" + 0.027*"solid_fuel" + 0.023*"pneumonia" + "0.016*"crack" + 0.012*"black_carbon"')] | Fossil Fuel Combustion |

stubble burning, excessive fossil fuel combustion, and construction activities should be effectively banned (0.009). Air pollution in the city of Mumbai (0.019) is represented as a serious concern. To deal with this, people (0.008) need (0.015) to ensure safe practices to achieve clean (0.014) air quality in the area.

The seventh set of words points to the use of fire (0.089) crackers (0.096) and stubble (0.053) burning as significant sources of air pollution in India. Firecrackers (0.052) are frequently used in India to celebrate the religious (0.014) festival of Diwali. The combustion of highly toxic chemicals in the crackers (0.096) contributes to the rise in air pollution. Similarly, every year, crop waste referred to as stubble (0.053) is burnt, leading to a spike in air pollution during the winter. This topic highlights two practices: (1) bursting (0.031) fire (0.089) crackers (0.096) and (2) stubble (0.053) burning, which contribute to a high concentration of air pollutants in India.

The eighth set of keywords addresses the adverse impacts of air pollution on the quality of life. Exposure to toxic pollutants in the air damages the human eye's tear film (0.039). High concentrations of air pollutants impact plants (0.036) as the toxins in the air seep into the soil, which strips the land of nutritional content and disrupts the water (0.035) pressure. The worse air quality during the winter season causes hazardous smog and haze, resulting in the shutting down of schools (0.035) and work (0.035) sites (0.019). This, in turn, also impacts the labor market (0.016).

The ninth set of keywords indicates that in reality (0.097), the use of solid-fuel (0.027) for cooking (0.065) stoves (0.036) is a significant contributing factor towards increased black-carbon (0.012) emissions in the country. The emissions from household cooking (0.065) fuel increase indoor air pollution and the risk of developing diseases such as pneumonia (0.023). The word cloud of to topics extracted from the Indian corpus are shown in Fig 7.

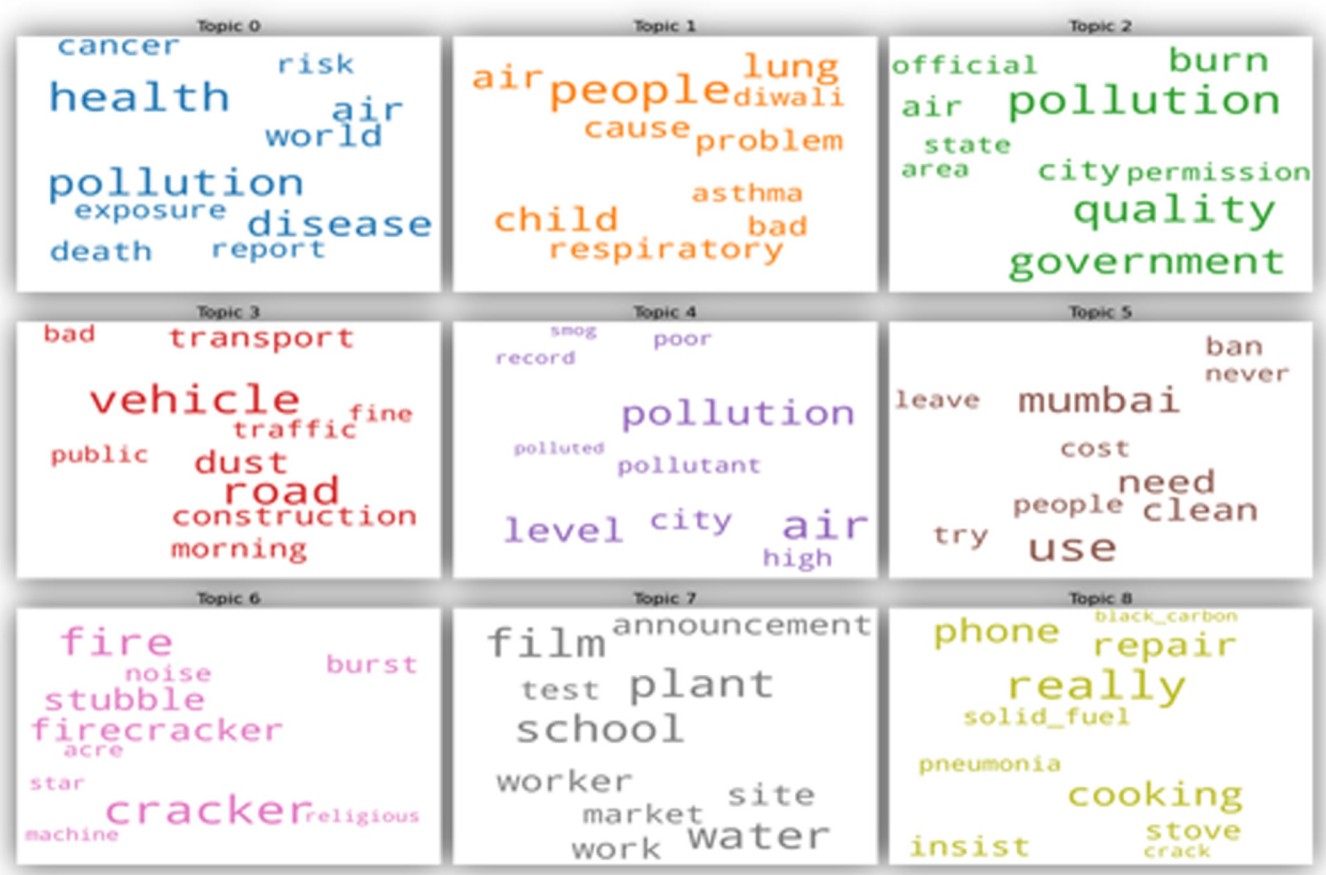

**Fig 7. Word cloud of India's topics extracted using LDA.**

A T-SNE distribution of topics on air pollution in India in Fig 8 is a visual representation of the nine topics identified related to air pollution in India. The figure uses T-SNE, a dimensionality reduction technique, to project high-dimensional data onto a 2D plane while preserving the relative distances between the data points. This way, the figure allows for easy visualization and comparison of India's topics. The figure can help us understand the main areas of focus related to air pollution in India.

Fig 9 visualized topics generated through LDA baed on Indian newspaper corpus. It shows the relevant topics. The topics that are closer or overlapping with each other show that they share characteristics on semantics bases. Moreover, the size of the topics in Fig 9 refers to the size of the topic in the corpus.

## 5 Discussion

The research used a robust methodology to examine the linguistic representation of air pollution in the newspaper discourses of two developing countries, Pakistan and India. The newspaper corpora for both countries were selected focusing air pollution dynamics from 2005 to 2023. The research employed LDA to uncover latent topics for both countries, providing a comprehensive understanding of the newspaper coverage of air pollution. The theoretical framework of Computational Grounded Theory (CGT), proposed by [NO_PRINTED_-FORM] [46],was used to guide the research in the detection of latent patterns (topics) in the

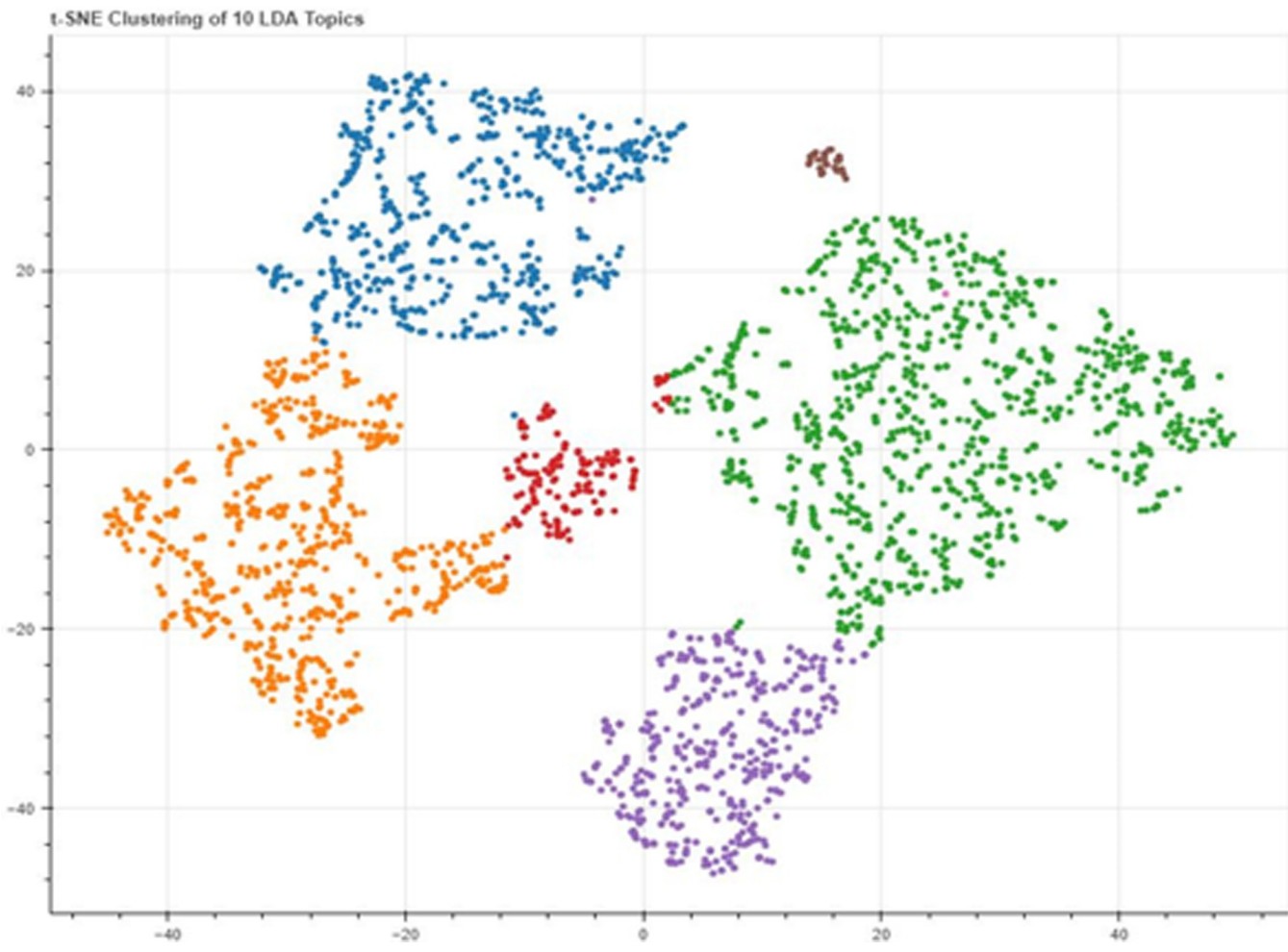

**Fig 8. A t-SNE distribution of topics extracted from the Indian Newspapers' corpus.**

corpus using LDA. These topics were then refined via computationally guided deep reading, and the inductively identified and refined topics were validated. A total of ten topics that emerged from the newspaper corpus of Pakistan are represented in Table 2. A total of nine topics emerged from the corpus of India, as represented in Table 3.

The research findings from Pakistan's newspaper-corpus, as shown in Table 2, reveal a diverse range of topics/discourses related to air pollution. These topics can be categorized into three main themes, each offering a unique perspective on air pollution dynamics. The first theme uncovers discourses regarding the toxicological effects of air pollution on human health, plants, and the environment in Pakistan. The discourses regarding the effects of air pollution on human health include death risk (Topic 9), respiratory ailments such as asthma and lung diseases (Topic 3), effects on human eyes (Topic 3), and decreased quality of life (Topic 1). According to the scientific data on the health effects of air pollution, air toxicants mainly contribute to the onset of respiratory [47], neuropsychiatric [48], cardiovascular [49], ophthalmologic [50], and oncological disorders [51]. In addition to affecting human health, air pollution also harms plants and the environment in Pakistan [17]. Air pollution damages agriculture (Topic 7), reduces crop yields and harms wildlife (Topic 7). The excessive reliance on traditional brick kilns in Pakistan also contributes to global warming and climate change (Topic 6)

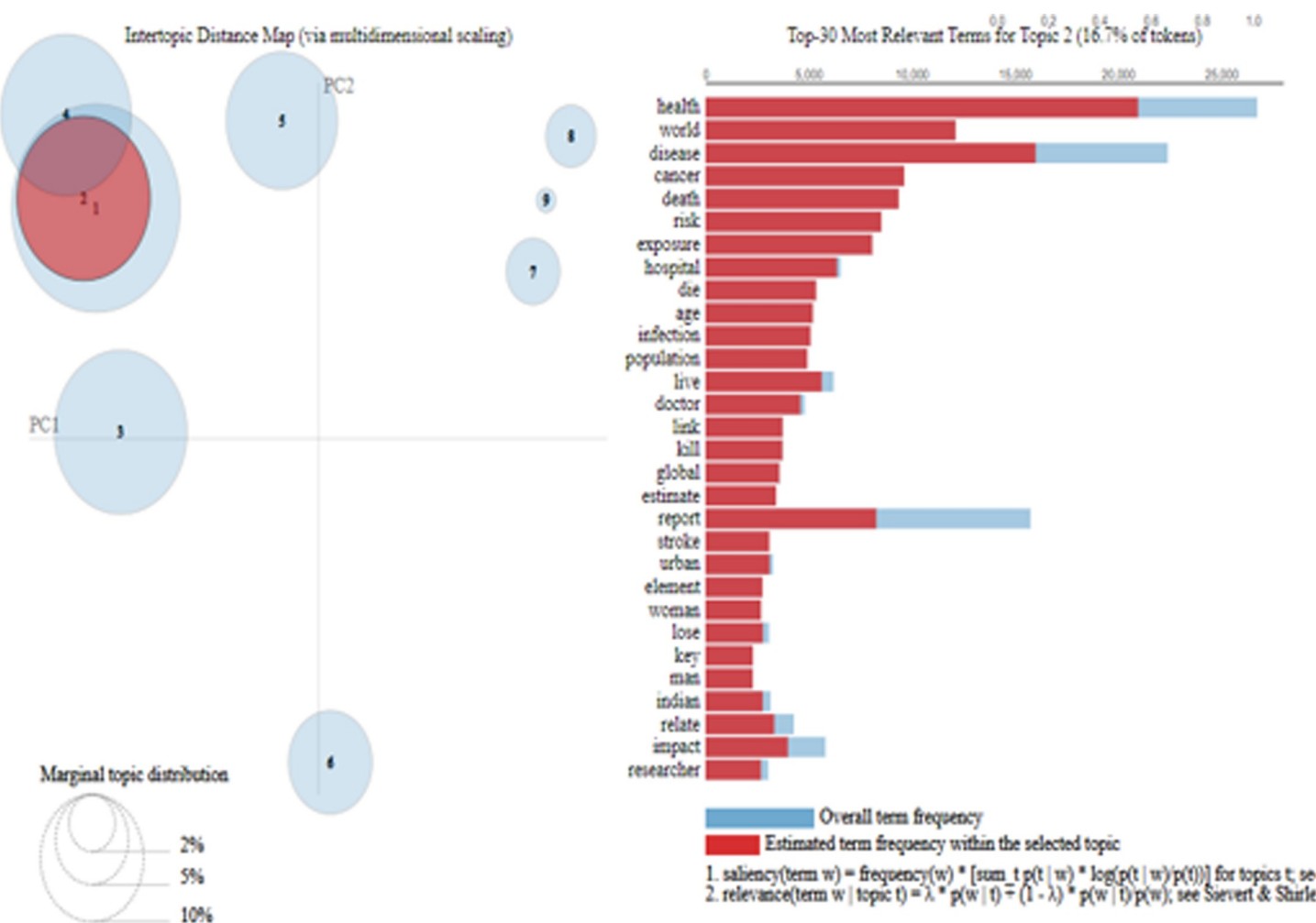

**Fig 9. LDA topic visualization India based on Indian Newspapers' corpus.**

through the release of greenhouse gases such as carbon dioxide and methane into the atmosphere, resulting in the melting of glaciers in the country's north, leading to more frequent and intense floods in the country [52]. Overall, air pollution can have a wide range of adverse effects on human health, plants, and the ecosystem.

The second theme relates to discourses based on factors which contribute to air pollutioin mainly transportation, industrial emissions (brick kilns, factories, and coal power plants), population growth, and urbanization. The discourses in newspapers regard vehicular emission as one of the primary sources of increased air pollution in Pakistan (Topic 2). Studies on the situation of air pollution in Pakistan indicate a direct association between vehicular emissions and air pollution through the release of toxic emissions such as carbon monoxide (CO), nitrogen oxides (NOx), volatile organic compounds (VOCs), and particulate matter (PM) [53,54]. Hence, there is a need to implement traffic mitigation measures in Pakistan from economic, technical, and administrative aspects to minimize vehicular emissions. In addition, the analysis of the generated topics reveals that industrial and domestic combustion of fossil fuels significantly contribute to air contamination (Topic 5). The thematic analysis also reveals that fossil fuel combustion in Pakistan has generated a concurrent air pollution crisis, primarily due to

the excessive reliance on traditional brick kilns utilizing fossil fuels spew innumerable suspended particulate matter into the atmosphere, causing high levels of air pollution (Topic 5). Scientists have established a wide range of significant health problems linked with using fossil fuels, as the toxins released from fossil fuel combustion contribute to highly unhealthy air quality [55,56]. Lastly, the topics that emerged on the contributing factors to air pollution in Pakistan (Topic 1, 4, Topic 7) indicate a strict correlation between air pollution, population growth, urbanization, and industrialization. The rapid population growth and construction activities have led to the ever-degrading air quality in Pakistan. It highlights a severe need on the part of the government to take action to reduce pollutant emissions from fossil fuel combustion.

The third thematic layer highlights the discourses based on policy measures taken by the Pakistani government over the past seventeen years to deal with the issue to overcome the menace of air pollution. The topics reveal some of the significant air pollution control policies and measures taken by the Pakistani government, including the National Clean Air Programme (NCAP) to reduce air pollution, stricter emission standards for industries and vehicles (Topic 2 & 8), and Pakistan Environmental Protection Agency (PEPA) to promote the use of clean energy especially in the province of Sindh (Topic 0).

The topics extracted from India's corpus shown in Table 3 can be categorized into three main thematic layers revealing the country's air pollution dynamics. The first thematic layer in India's topics reveals the toxic effects of air pollution on citizens' health. Significant health hazards related to air pollution in India are revealed to be death risk caused by fatal diseases such as cancer (Topic 0), respiratory diseases such as lung cancer and asthma (Topic 1), effects on skin and human eyes (Topic 7) and disturbance in life routine (Topic 7). Numerous studies in India have linked air pollution to the onset of respiratory diseases, cardiovascular dysfunctions, and cancer, making it a major death-causing risk factor in India [57,58]. Air pollution is also reported to cause increased levels of aggression and anxiety among individuals, leading to decreased quality of life in India (Topic 7). Also, severe damage is caused to human eyes by exposure to air pollution (Topic 7) in the form of lacrimation, corneal opacity, and dryness [4,59].

The second thematic layer relates to the contributing factors of air pollution in India. Significant sources of air pollution identified via topic modeling are transportation, industrial emissions, stubble burning, firecrackers, and urbanization. The analysis of discourses reveal two major contributing factors to air contamination are the use of excessive firecrackers during the religious festival of Diwali and stubble burning (Topic 7). In addition, vehicular emission from the transportation sector is identified as a source of increased air pollution in India (Topic 3). Moreover, industrial and domestic combustion of fossil fuels significantly contributes to air contamination in India (Topic 8). Fossil fuel combustion in India has generated a concurrent air pollution crisis, as the combustion releases enormous amounts of fine airborne particles (PM 2.5), sulfur dioxide, nitrogen oxides, polycyclic aromatic hydrocarbon (PAH), mercury, and volatile chemicals that form ground-level ozone [60]. Researches have also established a wide range of significant health problems linked with the use of fossil fuels as the toxins released from fossil fuel combustion contribute to highly unhealthy air quality [55].

The third thematic layer relates to the debate on policy measures India took to mitigate air pollution. The topics that emerged on policy measures taken by India revolve around initiatives to ensure improved quality of air in cities (Topic 4), control on methane emissions coming from stubble burning, and efforts towards effective implementation of the ban on industrial sources of emissions in the country (Topic 5). However, there is a lack of attention given to the emissions from the transport sector which is one of the primary contributing factors to air pollution in the country (Topic 3).

Table 4 compares the health hazards, sources, and air pollution policies in Pakistan and India.

Table 4 explains that all the thematic analysis of the topics, obtained through LDA, reveals that at level of heath hazrads the challenges are same in Pakistan and India. However, the sources for these health hazards differ. Urbanization and traditional brick kilns are distinctive challenges for Pakistan, whereas construction emissions, firecrackers burning and stubble burning are specific to Indian newspaper corpus. The common sources for air pollution are vehicular and industrial emissions.

## 6 Conclusion

Based on the analysis and discussion of the identified topics on air pollution from Pakistan and India corpus, the study concludes that air pollution has emerged as a significant issue for nearly the last two decades in the neighboring countries, with severe repercussions for human health and the ecology leading to high morbidities and mortalities. The discourses also reveal that the sources of air pollution are emerging from the potent combination of industrialization, urban development, and mass consumption in Pakistan and India.

The study has implications for policymakers as they can play a vital role in dealing with the threat of pollution in both countries. It proposes reforms in energy sector to reduce and control emssions by implementing strict emission standard for industries and vehicles. The focus on sustainale development and project can bring fruitful results in controlling pollution. This can include policies on smooth transition from fossil fuel to renewable energy. Similarly policymakers may impose ban on the sale of firecrackers during the certain times of the year and upgrade the brick kilns zigzag technology. For this purpose, a robust environmental protection organization must regulate the activities of different contributing factors to air pollution in the country.

In addition, given the transboundary nature of the air pollution crisis, Pakistan and India need to coordinate at the subnational, national, and regional levels to better achieve the desired goals. While the governments may take the required steps to enhance air quality, public collaboration and support are also critical to successfully implementing government policies. To improve the nation's air quality, all segments of society must fulfill their ethical and social duties. Hence, a joint strategy to combat the menace of air pollution in both countries may improve the air quality index in many cities of both neighbors.

The present study also has limitations. The first limitation is that it only focused on newspaper discourses reported in newspapers. Hence, the study relied on the corpus, which led to a theoretical study instead of an experimental one. The hazardous effects of air pollution can

**Table 4. Comparison of air pollution dynamics in Pakistan and India.**

|  | Health Hazards | Sources | Control Measures |
|---|---|---|---|
| **Pakistan** | Death Risk | Vehicular Emissions | Emission Standards for the Transportation Sector and Industries |
|  | Respiratory Ailments | Industrial Emissions |  |
|  | Reduced Quality of Life | Urbanization and Population Growth | Clean Power Generation Plants |
|  | Effects on the Human Eye | Traditional Brick Kilns |  |
| **India** | Death Risk | Vehicular Emission | Urban Pollution Control Measures |
|  | Respiratory Ailments | Construction Emission |  |
|  | Reduced Quality of Life | Industrial Emission | Measures to Control Industrial Emissions |
|  | Effects on the Human Eye | Firecrackers Burning |  |
|  |  | Stubble Burning | Measures Against Stubble Burning |

also be seen through the data based on facts from medical science to investigate the exact magnitude and intensity of the pollution in both countries. Future research based on empirical evidence may provide a better picture of on-ground reality in both countries.

Another limitation is that newspaper discourses may have biases in reporting events vis-à-vis their agenda-setting, which might have affected, to some extent, the impartial view of the matters reported in newspapers. Future research may be conducted by using the discourses from environmental experts' views or research reports to investigate air pollution.

## Author Contributions

**Conceptualization:** Fasih Ahmed.

**Data curation:** Fasih Ahmed.

**Formal analysis:** Sana Rabbani.

**Supervision:** Fasih Ahmed.

**Visualization:** Fasih Ahmed.

**Writing – original draft:** Sana Rabbani.

**Writing – review & editing:** Sana Rabbani.

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
