## [Decision Letter · Decision Letter 0]

23 Jul 2024

PONE-D-24-13028A Computational Analysis of Air Pollution Discourses in English Print Media of India and PakistanPLOS ONE

Dear Dr. Ahmed,

Thank you for submitting your manuscript to PLOS ONE. After careful consideration, we feel that it has merit but does not fully meet PLOS ONE’s publication criteria as it currently stands. Therefore, we invite you to submit a revised version of the manuscript that addresses the points raised during the review process.

We look forward to receiving your revised manuscript.

Kind regards,

Akhtar Malik Muhammad, PhD, Postdoc

Academic Editor

PLOS ONE

Journal Requirements:

5. Please ensure that you refer to Figure 2, 3, 4, 5 and 7 in your text as, if accepted, production will need this reference to link the reader to the figure.

**Additional Editor Comments:**

Dear Authors,

Thank you for submitting the revised manuscript " A Computational Analysis of Air Pollution Discourses in English Print Media of India and Pakistan " to PLOS ONE.

The reviewers have not recommended your paper and suggest major/minor revisions. The reviewer’s comments need to be addressed to improve the paper quality. I go through the revised manuscript and reviewers’ comments. I suggest you should address the reviewer’s comments and resubmit a revised version.

Yours sincerely

Dr. Malik Muhammad Akhtar

Academic Editor

PLOS ONE

Reviewers' comments:

Reviewer's Responses to Questions

**Comments to the Author**

1. Is the manuscript technically sound, and do the data support the conclusions?

Reviewer #1: Yes

Reviewer #2: Yes

Reviewer #3: Yes

2. Has the statistical analysis been performed appropriately and rigorously? 

Reviewer #1: Yes

Reviewer #2: Yes

Reviewer #3: Yes

3. Have the authors made all data underlying the findings in their manuscript fully available?

Reviewer #1: No

Reviewer #2: No

Reviewer #3: Yes

4. Is the manuscript presented in an intelligible fashion and written in standard English?

Reviewer #1: Yes

Reviewer #2: Yes

Reviewer #3: Yes

5. Review Comments to the Author

Reviewer #1: This is a very relevant work using LDA on newspaper reports pertaining to air pollution in India and Pakistan. I suggest the following minor changes to improve the paper.

In the methodology section, add which tools/packages were used for analysis.

In the methodology section, add which city edition you used for each newspaper.

In the discussion section, add some more discussion on Table 4, citing relevant studies.

In conclusion section, explicitly add recommendations for policymakers.

In conclusion section, explicitly add limitations and future work.

Proof Corrections:

In paragraph 2 of Introduction, one reference (Kumar & Omidvarborna, 2018) is not part of any sentence.

In Literature Review, one reference (Nasir et al) does not have year. If no information is available add (n.d.)

I wish you best for revisions.

Reviewer #2: Methodology: The authors have used Latent Dirichlet Allocation (LDA), a topic modeling technique, to analyze the newspaper discourse on air pollution in India and Pakistan. However, more details on the preprocessing of the text data, the number of topics chosen, and the evaluation of the topic model would be helpful to assess the robustness of the methodology.

Scope and Timeframe: The study focuses on newspaper discourses from 2005-2023, which is a reasonable timeframe to capture the evolution of the air pollution discourse in the two countries. However, it would be interesting to know if the authors considered any specific events or policy changes that may have influenced the discourse during this period.

Data Sources: The authors mention analyzing English print media from India and Pakistan, but more details on the specific newspapers, circulation, and readership would provide better context for the analysis.

Topic Modeling: The authors have used LDA to extract the primary topics related to air pollution discourses. It would be helpful to know if they explored alternative topic modeling techniques, such as Non-Negative Matrix Factorization (NMF) or Hierarchical Dirichlet Process (HDP), to compare the results and ensure the robustness of the findings.

Topic Interpretation: The authors have provided interpretations of the key topics identified by the LDA model. It would be useful to know how they validated these interpretations, such as through expert interviews or comparison with existing literature.

Cross-country Comparison: The study compares the air pollution discourses in India and Pakistan. It would be interesting to know if the authors found any significant differences in the topics or their relative importance between the two countries, and what might be the underlying reasons for such differences.

Temporal Trends: The study covers a relatively long timeframe of almost two decades. It would be valuable to understand if the authors observed any significant changes in the air pollution discourses over time, and how these changes might be linked to specific events or policy interventions.

Policy Implications: The authors mention that the study has implications for policymakers and planners. It would be helpful to elaborate on the specific policy recommendations that emerge from the analysis, and how they can be used to address the air pollution challenges in the two countries.

Limitations and Future Research: The authors could discuss the limitations of the study, such as the reliance on English-language media or the potential biases in newspaper reporting, and suggest avenues for future research to build upon this work.

Contextual Factors: Air pollution discourse is likely influenced by a range of contextual factors, such as economic development, urbanization, and political dynamics. It would be interesting to understand how the authors accounted for these factors in their analysis or how they could be incorporated in future research.

Comparison with Other Studies: The authors could situate their findings within the broader literature on air pollution discourses, both in the South Asian region and globally, to highlight the unique contributions and insights of their study.

Qualitative Analysis: While the study relies primarily on computational techniques, a complementary qualitative analysis of the newspaper articles could provide additional depth and nuance to the understanding of air pollution discourses.

Stakeholder Perspectives: The study focuses on media discourses, but it could be valuable to explore the perspectives of other key stakeholders, such as policymakers, industry representatives, and civil society organizations, to gain a more comprehensive understanding of the air pollution issue.

Implications for Public Engagement: The findings of the study could have implications for how air pollution is communicated to the public and how citizens can be engaged in addressing this challenge. The authors could discuss these aspects and their potential impact.

Broader Societal Impacts: Air pollution has significant consequences for public health, the environment, and economic development. The authors could discuss how their findings relate to these broader societal impacts and how the insights from the study could contribute to more effective air pollution mitigation strategies.

Reviewer #3: Thank you for the inteligent piece about the topic. The study is interesting. I hvae a few suggestions to make:

1. In the Introduction, please avoid the repetition around the literature under the Literature Review heading. Fisrt para has significant amount of repeated information from the previous section. You may want to break the subseuqent information sub-heading wise that will make the reader understand the topic deeply. Omit 'Literature Review' heading and break the section sub-heading wise.

2. Results have been presented well, however, it will be good if you can prepare a comparative discourse for the two countries and build the discussion accordingly. Individual details are fine but a comparative will largely help.

3. Discussion can be refined in a more scientific manner which can follow a strategic step-wise discussion that may involve firstly the studies of such kind that are nationally and globally available and a comparatie thereforth; secondly the results can be compared with other relevant studies done in developing countries that may or may not have used the similar approach; thirdly, it will good to also compare how the two discourses are different and what it may imply; fourthly, the conclusion can also talk about the possible actions country-wise based on the respective results and also how media reportings are relevant and make an impact on such issues and how can they be improved.

Thanks

6. PLOS authors have the option to publish the peer review history of their article (what does this mean?). If published, this will include your full peer review and any attached files.

Reviewer #1: **Yes: **Deepak Saxena

Reviewer #2: No

Reviewer #3: **Yes: **Dr Priyanka Rani Garg

---

## [Author Response · Author response to Decision Letter 0]

23 Sep 2024

Response to the Reviewers

 Comments Responses

Reviewer 1 In the methodology section, add which tools/packages were used for analysis. 

Thank you very much for pointing out regarding the mentioning of tools/packages. We have mentioned side by side with in methodology section the detail regarding tools/packges used for preprocessing, data filteration, and topic modeling (please see page 9, 10, and 11 )

 In the methodology section, add which city edition you used for each newspaper Thank you very much for pointing out regarding the circulation of the newspapers. We developed criteria based on the following five points: 

a) daily publication

b) universal national coverage

c) large circulation

d) reputable journalistic standards

e) highest news index on air pollution.

(Please see page 9 and 10 section sampling)

 In the discussion section, add some more discussion on Table 4, citing relevant studies. Thank you for pointing out regarding the improvement in the discussion section. The changes have been incorportated and the more relevant studies have been cited (please see page 25,26 and 27 section discussion)

 In conclusion section, explicitly add recommendations for policymakers. Conclusion section has been improved. 

 In conclusion section, explicitly add limitations and future work. Limitations and future work suggestions have been included in the study. 

Reviewer 2 The authors have used Latent Dirichlet Allocation (LDA), a topic modeling technique, to

analyze the newspaper discourse on air pollution in India and Pakistan. However, more details on the preprocessing of the

text data, the number of topics chosen, and the evaluation of the topic model would be helpful to assess the robustness of

the methodology. Thank you very much for pointing out to incorporate the details regarding the evaluation of the model. A complete detail regarding how the model has been refined and applie has been incorporated (please see page 9, section LDA)

 The study focuses on newspaper discourses from 2005-2023, which is a reasonable timeframe to

capture the evolution of the air pollution discourse in the two countries. However, it would be interesting to know if the

authors considered any specific events or policy changes that may have influenced the discourse during this period. The study, as explained in the introductory section, focused only on newspaper discourses from 2005 to 2023. The analysis with respect to specific events or policy analysis has not been conducted. This has been done for the two main reasons;

a) Firstly, the study used computational Grounded theory with no prior objectives or research questions. 

b) Secondly, newspaper discourses may not be a suitable source for investigating the policy changes. 

 Data Sources: The authors mention analyzing English print media from India and Pakistan, but more details on the specific newspapers, circulation, and readership would provide better context for the analysis. The detail regarding the newspaper circulation has been incorporated to better explain the context of the circulation of newspapers in both countries. 

 The authors have used LDA to extract the primary topics related to air pollution discourses. It would be

helpful to know if they explored alternative topic modeling techniques, such as Non-Negative Matrix Factorization (NMF) or

Hierarchical Dirichlet Process (HDP), to compare the results and ensure the robustness of the findings The study was delimited to LDA to extract topics as compared to the alternative topic modeling techniques. In case of making a comparison of techniques, it would become quite complex and in that case it may reflect to validate which topic modeling method is comparative better as compared to other which is not the objective of the study. Hence keeping in view that point, the study was only delimited to LDA topic modeling. 

 The study compares the air pollution discourses in India and Pakistan. It would be interesting to know if the authors found any significant differences in the topics or their relative importance between the two

countries, and what might be the underlying reasons for such differences. The study examines the dynamics of air pollution in Pakistan and India. The analysis has been conducted separately for comparison. Moreover, a table has been added in the discussion section that explains the difference in topics (see page 33 Table 4 section discussion)

 The study covers a relatively long timeframe of almost two decades. It would be valuable to understand if the authors observed any significant changes in the air pollution discourses over time, and how these changes might be linked to specific events or policy interventions. The rationale for covering the long time frame has been mentioned in the introduction section. The study's main objective is to cover the hazards of air pollution. Hence, the study was delimited to only newspaper discourses, irrespective of specific events. However, the rationale for time-frame selection has been provided in the introduction section of the article (see pages 2, 3, and 4).

 The authors mention that the study has implications for policymakers and planners. It would be helpful to elaborate on the specific policy recommendations that emerge from the analysis and how they can be used to address the air pollution challenges in the two countries The conclusion section includes a separate paragraph for implications and future suggestions (see pages 34 and 35, section conclusion).

 The authors could discuss the limitations of the study, such as the reliance on English language media or the potential biases in newspaper reporting, and suggest avenues for future research to build upon this work. Thank you very much the suggestions have been compiled (see page 35, section conclusion)

 Air pollution discourse is likely influenced by a range of contextual factors, such as economic development, urbanization, and political dynamics. It would be interesting to understand how the authors accounted for these factors in their analysis or how they could be incorporated in future research. As explained earlier, the study is not context-specific. Hence, we have not considered the contextual factors contributing to air pollution. However, we have mentioned this in the limitations of the study (see page 34, 35 and 36 section conclusion). 

 While the study relies primarily on computational techniques, a complementary qualitative analysis of the newspaper articles could provide additional depth and nuance to the understanding of air pollution discourses. Thank you very much for pointing out the gap in the study. Though the main part of the study is quantitative, we have underpinned a Computational Grounded Theory, where a mixed-method research design has been applied. After obtaining the results quantitatively, we have analyzed them qualitatively. 

 The study focuses on media discourses, but it could be valuable to explore the perspectives of other key stakeholders, such as policymakers, industry representatives, and civil society organizations, to gain a more comprehensive understanding of the air pollution issue. We agree with the suggestion, but if the study is extended to that level, it may lose its present scope. However, we have mentioned the limitations and future research suggestions. In this way, it may be indicative for other researchers to carry forward the work from this point onward. 

Reviewer 3 In the Introduction, please avoid the repetition around the literature under the Literature Review heading. Fisrt para has significant amount of repeated information from the previous section. You may want to break the subseuqent information sub-heading wise that will make the reader understand the topic deeply. Omit 'Literature Review' heading and break the section sub-heading wise. Thank you for pointing out the issue. The redundant information in the literature review section has been omitted. 

In addition, literature review section has been divided into subheading as per the recommendations (see pages 6, 7 and 8 section literature review).

 Results have been presented well; however, it will be good if you can prepare a comparative discourse for the two countries and build the discussion accordingly. Individual details are fine but a comparative will largely help. The comparative discourse has been incorporated (see page 34 Tabe 4) 

 Discussion can be refined in a more scientific manner which can follow a strategic step-wise discussion that may involve firstly the studies of such kind that are nationally and globally available and a comparatie thereforth; secondly the results can be compared with other relevant studies done in developing countries that may or may not have used the similar approach; 

 Thank you for your comments to improve discussion section. It have been thoroughly revised. In addition a comparative analysis has been incorporated (see pages 29-34 section discussion). 

 It will good to also compare how the two discourses are different and what it may imply. The comparison of the results has been thorugh conducted in the discussion section A separate table regarding the comparative analysis has been included (see page 34, Table 4)

 Conclusion can also talk about the possible actions country-wise based on the respective results and also how media reportings are relevant and make an impact on such issues and how can they be improved. The conclusion section of the study has been revision as per the recommendations (see pages 34, 35).

---

## [Decision Letter · Decision Letter 1]

21 Nov 2024

PONE-D-24-13028R1

A Computational Analysis of Air Pollution Discourses in English Print Media of India and Pakistan

PLOS ONE

Dear Authors,

I am pleased to inform you that your manuscript has been accepted for publication. Congratulations on your accomplishment, and thank you for choosing *PLOS ONE* as the venue to share your research.

The next steps in the publication process will be managed by the production team.

Best regards,

Syed Ahsan Ali Shah

Academic Editor

*PLOS*
*ONE*

---

## [Editor Report · Acceptance letter]

28 Nov 2024

PONE-D-24-13028R1 

PLOS ONE

Dear Dr. Ahmed, 

I'm pleased to inform you that your manuscript has been deemed suitable for publication in PLOS ONE. Congratulations! Your manuscript is now being handed over to our production team.

Kind regards, 

on behalf of

Dr. Syed Ahsan Ali Shah 

Academic Editor

PLOS ONE